# Kinetic Models of Wealth Distribution with Extreme Inequality: Numerical Study of Their Stability against Random Exchanges

**DOI:** 10.3390/e25071105

**Published:** 2023-07-24

**Authors:** Asim Ghosh, Suchismita Banerjee, Sanchari Goswami, Manipushpak Mitra, Bikas K. Chakrabarti

**Affiliations:** 1Department of Physics, Raghunathpur College, Raghunathpur, Purulia 723133, India; 2Economic Research Unit, Indian Statistical Institute, Kolkata 700108, India; suchib.1993@gmail.com (S.B.); manipushpak.mitra@gmail.com (M.M.); bikask.chakrabarti.retd@saha.ac.in (B.K.C.); 3Department of Physics, Vidyasagar College, Kolkata 700006, India; sg.phys.caluniv@gmail.com; 4Saha Institute of Nuclear Physics, Kolkata 700064, India

**Keywords:** wealth inequality, kinetic exchange models, Yard-Sale model, Monte Carlo simulations

## Abstract

In view of some recent reports on global wealth inequality, where a small number (often a handful) of people own more wealth than 50% of the world’s population, we explored if kinetic exchange models of markets could ever capture features where a significant fraction of wealth can concentrate in the hands of a few as the market size *N* approaches infinity. One existing example of such a kinetic exchange model is the Chakraborti or Yard-Sale model; in the absence of tax redistribution, etc., all wealth ultimately condenses into the hands of a single individual (for any value of *N*), and the market dynamics stop. With tax redistribution, etc., steady-state dynamics are shown to have remarkable applicability in many cases in our extremely unequal world. We show that another kinetic exchange model (called the Banerjee model) has intriguing intrinsic dynamics, where only ten rich traders or agents possess about 99.98% of the total wealth in the steady state (without any tax, etc., like external manipulation) for any large *N* value. We will discuss the statistical features of this model using Monte Carlo simulations. We will also demonstrate that if each trader has a non-zero probability *f* of engaging in random exchanges, then these condensations of wealth (e.g., 100% in the hand of one agent in the Chakraborti model, or about 99.98% in the hands of ten agents in the Banerjee model) disappear in the large *N* limit. Moreover, due to the built-in possibility of random exchange dynamics in the earlier proposed Goswami–Sen model, where the exchange probability decreases with the inverse power of the wealth difference between trading pairs, one does not see any wealth condensation phenomena. In this paper, we explore these aspects of statistics of these intriguing models.

## 1. Introduction

The first successful theory involving classical many-body physics or classical condensed matter systems, the kinetic theory of the (classical) ideal gas, is about one-and-a-quarter centuries old. It consists of Avogadro’s number (about $10^{23}$) of constituent atoms or molecules (each following Newtonian dynamics). It is a robust, versatile, and extremely successful foundation of classical many-body physics. Social systems, economic markets in particular, are many-body dynamical systems composed of fewer constituents (ranging from the order of $10^{10}$ for a global market). A lone Robinson Crusoe on an island cannot develop a market or a society for that matter, as markets are intrinsically many-body systems. It is no wonder that kinetic exchange models of money or wealth have been conjectured early on (e.g., by Saha and Srivastava [1] in 1931, Mandelbrot [2] in 1960) and resurrected recently (e.g., by Chakrabarti and Marjit [3] in 1995, Dragulecu and Yakovenko [4] in 2000, Chakraborti and Chakrabarti [5] in 2000, Chatterjee, Chakrabarti, and Manna [6] in 2004).

Kinetic exchange models of trades and their statistics have been quite successful in capturing several realistic features of wealth distributions among agents in societies (see, e.g., [7,8]). The beneficial effects of an agent’s saving propensity in reducing social inequality have been extensively studied [5,6,8]. The choice of the poorest trader in each trade (with the other trade partner being randomly chosen) leads to a remarkable self-organized poverty line, beneath which, no one remains in a steady state (see, e.g., [9,10,11,12]). This model was inspired by some crucial observations by economists (see, e.g., [10]) and suggests built-in (self-organized) remedies for reducing social inequality. However, it must be admitted that such intriguing self-organizing properties of the kinetic exchange models have not yet been thoroughly investigated.

Contrarily, recent focus has shifted to the unusual growth rate of social inequality in the post world war II period (see, e.g., [13,14,15,16]), which in some countries seems to have significantly crossed the 80-20 Pareto limit, reaching a steady state, with 87% of the wealth accumulated by 13% of the population. This has been argued, following an analogy with the inequality index values for the avalanche burst statistics in self-organized sand-pile models near their critical points, to be the natural limit in all social competitive situations, where welfare mechanisms (helping those who fail to participate properly in such self-organizing dynamics) are either absent or removed (see, e.g., a recent review [16]).

Although the Pareto-like inequality mentioned above—where a small fraction of people (say 13%) possess a large fraction (say 87%) of wealth—can already be devastating, more disturbing types of inequalities are being reported. For example, the Oxfam Report [17] of January 2020 stated “The world’s 2153 billionaires have more wealth than the 4.6 billion people who makeup 60 percent of the planet’s population”. In other words, a handful number (about $10^{3}$) of rich people possess more than about 60% (or $10^{9}$ order) of poor people’s wealth on this planet. This dangerous trend in the world, as a whole, has repeatedly been mentioned in various recent reports.

The Pareto-type inequality mentioned above has long been investigated (see, e.g., [6,18]) using the kinetic exchange models with non-uniform saving propensities of traders (see, e.g., [8,19] for reviews). One may naturally wonder if the kinetic exchange theory allows for possible models, where only a handful of traders (say, about 10) possess a significant fraction (say, above 50%) of the total wealth considered in the model, even when its population *N* tends to infinity.

The answer is yes. The Chakraborti model [20], widely known today as the Yard-Sale model, as in [21], has attracted a lot of attention (see, e.g., [22,23,24]). In its barest form [20], in the Chakraborti model (denoted here as the C-model), two randomly chosen traders at any point in time participate in an exchange trade when the richer one saves the excess over the poorer one’s wealth and goes for a random exchange of the total available wealth (double that of the poorer one). The slow but inevitable attractor fixed point of the trade dynamics arrives when all wealth ends up in the hand of just one trader, no matter how large the population (*N*) is. Because of the particular form of savings during a trade, whenever one becomes a pauper, nobody trades with him, and all gradually condense to the state where one trader acquires the entire wealth and the trade dynamics stop (see [22]). External perturbations, like regular redistribution of tax collections by the central government (or any non-playing agent), can help relieve [23,24] the condensation phenomenon, and this seems to fit well with many observed situations [23]. We will show here that if each trader has a finite probability (*f*) of playing Dragulecu and Yakovenko (DY)-type [4] random exchanges, then for any f>0, the condensation of wealth in the hands of one trader disappears and the steady-state distribution of wealth exponentially decreases, as in the DY model.

In the Goswami–Sen (or GS) model [25], one considers a kinetic exchange mechanism, where the interaction (trade) probabilities among the trade partners (*i* and *j*) decrease with the wealth differences ($|m_{i}-m_{j}|$) at that instant of trading (time), following a power law ($|m_{i}-m_{j}{|}^{-\alpha }$). Of course, for α = 0, the model reduces to that of DY. The numerical results showed that, for α values of less than about 2.0, the steady-state wealth distribution among the traders was still DY-like (it exponentially decayed with increasing wealth). For higher values (beyond 2.0) of α, power law (Pareto-law) decays occurred. No condensation of wealth in the hands of a finite number of traders or agents was observed due to the inherent DY-like exchange probability in the dynamics of the model. This was confirmed by extrapolating the fraction of total wealth held in the steady state by the ten richest traders, with respect to *N*.

We finally consider a seemingly natural version of the kinetic exchange model, denoted here as the Banerjee (B) model [26], where the intrinsic dynamics of the model lead to another extreme kind of inequality in the steady state, in the sense that precisely ten traders (out of the *N* traders in the market; N→∞) possess (99.98 ± 0.01)% of the total wealth. These fortunate traders are not unique and their fortune does not last for long (the residence time, on average, is about 66 time units, with the most probable value around 25 time units, counted in units of *N* trades or exchanges, for any value of *N*) and it decreases continuously with the increasing fraction (*f*) of random trades or interactions. Unlike in the Chakraborti or Yard-Sale model [20,21], where the dynamics stop after the entire wealth goes to one (unless perturbed externally), the trade dynamics continue here with the total wealth circulating only with a handful of traders (about ten) in the steady state. In this model, after each trade, the traders are ordered from the lowest wealth to the highest, and each trader trades only with the nearest-in-wealth trader, richer or poorer, with equal probability. Even if, by chance, the entire wealth goes to one trader, the dynamics of random exchanges do not stop in this model as all the paupers become the nearest neighbors (wealth-wise) of this trader, and random exchanges among them occur. The process continues. Apart from the steady-state wealth distributions and the most probable wealth amounts of the top few rich traders, we will show that, in this model, the condensation of almost all the wealth (99.98%) occurs in the hands of 10 traders (no matter how big *N* is). We will show that this condensation disappears when a finite fraction *f* of the time traders engage in DY-type random exchanges. Eventually, a DY-type exponentially decaying wealth distribution emerges after a power law region for low values of *f*.

## 2. Models and Numerical Studies for Their Statistics

We numerically study the statistical features of the three kinetic exchange models introduced in the introduction. We begin with the B (Banerjee [26]) model. Next, we consider the C (Chakraborti, or Yard Sale) model [20,21], and then the GS (Goswami–Sen) model [25]. In order to explore the stability of the condensation of wealth in these models, we study the steady-state wealth distribution P(m) in each model and the fraction of total wealth concentrated in the hands of a few (say ten) traders or agents (whenever meaningful), allowing each trader to have a nonvanishing probability *f* (the faction of tradings or times) to go for DY (Dragulecu and Yakovenko [4])-type random exchanges.

Most of the numerical (Monte Carlo) studies of the dynamics of these models are studied with four sets of numbers *N* of agents or traders: *N* = 100, 200, 400, and 800, and at each time step, *t*, the dynamics run over all the *N* order traders. We consider total money (*M*) to be distributed among the agents equal to *N* and we denote the money of any agent *i* at time *t* by $m_{i}\left(t\right)$ and, as such, $M=\sum_{i}m_{i}\left(t\right)=N$. When the steady state is reached after the respective relaxation times, when the average quantities do not change with time (with the relaxation time typically being much less than $10^{5}$ trades/interactions for the *N* values considered here), the statistical quantities are evaluated from averages of over $10^{5}$ post-relaxation time steps.

### 2.1. Banerjee Model Results

In this B-model, when the DY fraction (*f*) is set equal to zero, no wealth distribution P(m) in the population is meaningful because of the wealth condensation in the hands of a few. We first study the distributions (see Figure 1) of the total wealth fraction in the hands of the richest three. Note that these three are not unique, and once they become so rich, their residence times (in units of *N*) are finite (about 66), and in case these positions are lost, the return times are also finite.

Although the distribution of the total wealth fraction in the hands of the richest few (shown in Figure 1) is rather wide (each one spread over more than 30% of the total wealth and not *N*), the distribution of the total wealth fraction possessed by the ten richest (at any time in the steady state) is extremely narrow and spreads over 0.1% only (see Figure 2). At any time in the steady state, its value is much more robust in this B model (with *f* = 0), and its value is less than unity, but very close to 0.9998.

Next, we consider the B model with a nonvanishing probability *f* of each trader to follow DY trades or exchanges. We see, immediately, that the wealth condensation disappears, and with increasing values of *f*, the wealth is Boltzmann (exponentially)-distributed among all the agents (see Figure 3), starting with the Pareto-like power law distribution for lower values of *f* (see the inset of Figure 3). Indeed, when we consider the limiting values (for large *N*) of the average fraction of total wealth (M=N) possessed by the ten richest traders in the steady state, they all seem to vanish (see Figure 4) for any non-zero value of *f* (there remains a constant of 0.9998 for *f* = 0, for the pure B model).

For the wealth condensation in the B-model (with *f* = 0), Figure 5A shows the distribution of residence times (in units of *N*) of the 10 fortunate traders and, in the inset, the variation of the most probable and average values of residence times (τ, in the unit of *N*). For the same model with *f* = 0, Figure 5B shows the distribution of the return time to fortune (becoming one of the 10 richest starts from the 20th rank) and (in the inset) the variation of the most probable and average values of the residence times with market sizes *N*.

### 2.2. Chakraborti or Yard-Sale Model Results

The C model or Yard sale model is well-studied. However, in order to check the stability of the condensation of wealth (with the entire money M=N going to the hands of one trader only, we added a nonvanishing probability *f* of each trader to follow DY trades or exchanges. We can immediately see that the wealth condensation disappears for any f>0 (see Figure 6) and the wealth is distributed in the Boltzmann form (exponentially decaying with increasing wealth) among all the agents. The inset shows that for any nonzero value of *f*, the steady state wealth distribution is exponentially decaying (and there is a power law region) in this extended model C. When we consider the limiting values (for large *N*) of the average fraction of total wealth (M=N) possessed by the ten richest traders in the steady state (see Figure 7), they all seem to vanish from the unit value in the original C model (with f=0) for any non-zero value of *f*.

### 2.3. Goswami–Sen Model Results

Here, the interaction (trade) probability (*i* and *j*) decreases with the wealth difference (|mi−mj|) at the instance of trading (time), following a power law (${\left|mi-mj\right|}^{-\alpha }$). As such, in the GS model, there is always a finite (but small) probability of random exchanges. We do not need to consider the additional fraction of the DY interaction in this model. Of course, for α = 0, the model reduces to that of DY. Our numerical results confirm (see Figure 8) that for α values of less than about 2.0, the steady-state wealth distribution is still DY-like (exponentially decaying). For higher values (beyond 2.0) of α, power law (Pareto-like) decays occur (but no condensation of wealth). Although the model leads to extreme inequality, there is no condensation of wealth in the hands of a few traders for any (larger) value of α. In order to check that, we studied the average fraction of total wealth (M=N) possessed by the ten richest traders in the steady state of the GS model with α. When we plot the fraction against $1/N^{2}$ (see Figure 9), the extrapolated values of the fraction all seem to approach zero for any non-zero value for any of the α values considered.

## 3. Summary and Discussion

In view of the observed extreme income or wealth inequalities in society, we investigated the suitability of capturing the kinetic exchange models [8], at least qualitatively. We distinguish between two types of such extreme inequalities: the (Pareto) type [16], where a small fraction (typically 13%) of the population possesses about 87% of the total wealth (following a power law distribution) of the respective country. The other more recently observed type (as reported by Oxfam [17]) regards the truly extreme nature of income and wealth inequalities worldwide, where only a handful (say a few hundred to thousands) of super-rich people throughout the world own more wealth than 50 to 60% of poor people.

Several kinetic exchange models (see, e.g., [6,8]) have been developed to analyze Pareto-type inequalities. We investigated the statistics of some kinetic exchange models, where even with *N* going to the infinity limit, only one person can grab the entire wealth (as in the Yard-Sale, Chakraborti, or C models [20,21]), or only 10 people can accumulate about 99.98% of the total wealth (as in the Banerjee or B model [26], see Figure 2). We investigate how these extreme inequalities in the kinetic models are softened to the Dragulescu–Yakovenko (DY) [4] types of exponentially decaying wealth distributions among all traders or agents, when the traders each have non-vanishing probabilities *f* of DY-type random exchanges. These condensations of wealth (100% in the hands of one agent in the C model [20], or about 99.98% in the hands of ten agents in the B model) then disappear in the large *N* limit (this is clearly seen when extrapolating against $1/N^{2}$, as in DY-type random exchanges, where each *N* agent interacts or exchanges with all others; see Figure 4 and Figure 7). We also show that due to the built-in possibility of DY-type random exchange dynamics in the Goswami–Sen or GS model [25], where the exchange probability decreases with the inverse power of the wealth difference of the pair of traders, one does not see any wealth condensation phenomena. In both GS and B models (with f>0 fraction DY interactions or exchanges) no wealth condensation occurs, although a strong Pareto-type power law wealth distribution P(m) or inequalities occur for large values of α, as well as smaller values of *f* in the GS and B models, respectively (see Figure 3 and Figure 8). For the wealth condensation in the B model, for *f* = 0, we additionally find that the top ten fortunate traders are not unique and their fortunes do not last for long (the residence time τ of the fortune, on average, is about 66 time units, with the most probable value being around 25 time units, when counted in units of *N* trades or exchanges; see Figure 5A). The most probable ‘return time’ to such a fortune (of the 20th rank holder, coming within the group of fortunate 10), is found to be about 20 (in units of *N*; see Figure 5B). It should be noted that with f=0, in the C-model, the residence time τ is infinity for the only fortunate one accumulating the entire wealth in the system. Indeed, with increasing values of DY fraction *f*, the values of τ in both cases decrease rapidly (see Figure 10), following inverse power laws with *f*. We further note that, for *f* = 0 in the B-model near the most probable values of the wealth fractions (Figure 1 and Figure 2) and residence or return times (Figure 5), the fluctuations tend to grow with *N*, indicating the possible divergence there in the macroscopic limit of *N*. We plan to explore this significance later.

Our studies for the B, C, and GS kinetic exchange models using Monte Carlo techniques suggest that the potential condensation type of extreme inequality can disappear in all of them if a non-vanishing probability of random exchange is allowed, converging to the Pareto-type power law inequality (for the B and GS models), converging to the Gibbs-like (exponentially decaying) wealth distribution for larger values of *f* in the B model, smaller values of α in the GS model, or any values of f>0 in the C model. These observations may help to formulate public welfare policies.

## Figures and Tables

**Figure 1 entropy-25-01105-f001:**
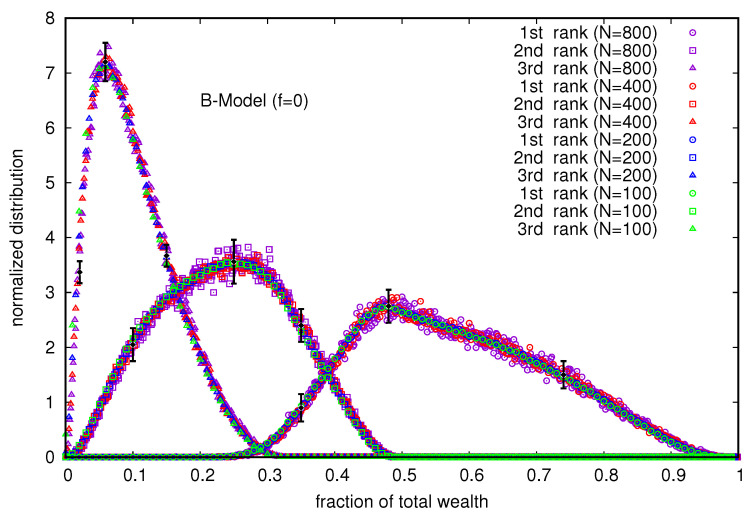
Distributions of the fraction of total wealth (M=N) ending up in the hands of the richest three traders. The error estimation is based on 10 runs. The typical errors in the distribution grow with *N* near the most probable value of the wealth fraction and are indicated for *N* = 800 for all three traders. Far away from the most probable values, the errors are less than the data point symbol sizes.

**Figure 2 entropy-25-01105-f002:**
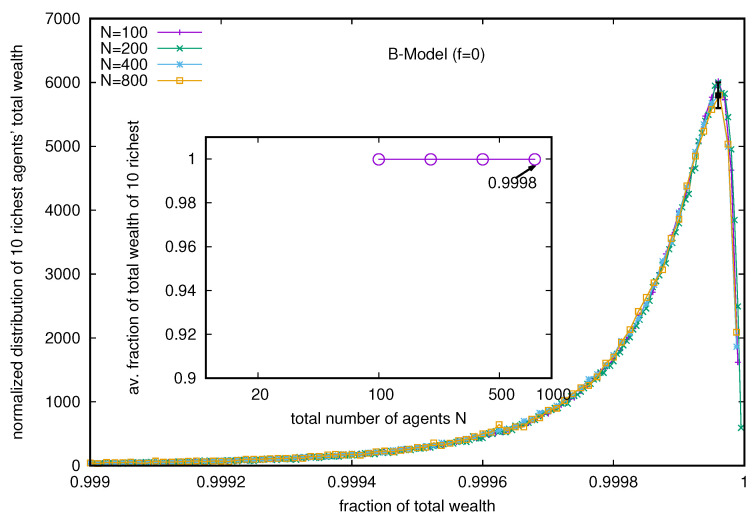
Distribution of the total wealth fraction possessed by the ten richest (at any time in the steady state and for different *N* values). The inset shows that the average of the total wealth fraction of the ten richest (for any time and any value of *N*) in the steady state is very close to 0.9998. Although the wealth share fractions of the richest ten traders have considerable fluctuations (see Figure 1), their wealth fraction totals hardly have any fluctuations (much less than the symbol size in the inset). The error estimation is based on 10 runs. The typical errors in the distribution of total wealth of the ten richest are more than the data point symbol sizes near the most probable values, where indicated.

**Figure 3 entropy-25-01105-f003:**
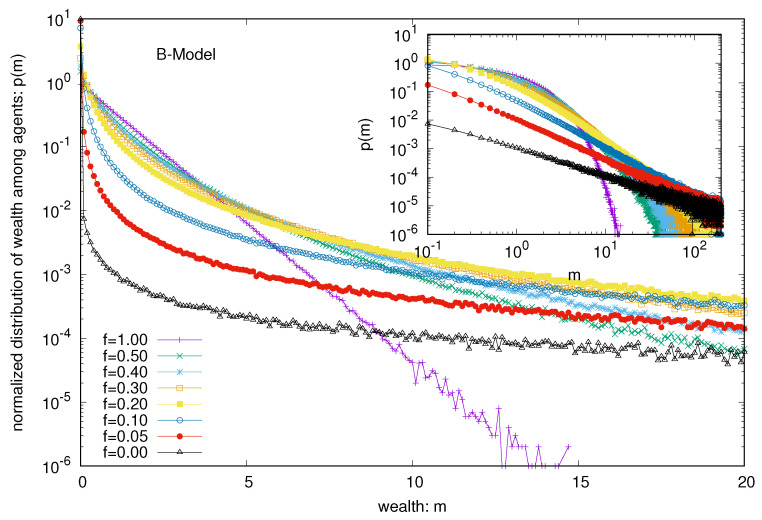
Wealth distribution P(m) among all agents against the wealth *m* in the B model for different probabilities *f* of DY random exchanges. Note that the fluctuations appear to grow more for the lower values of the distribution of wealth due to the log scale used in the y-axis.

**Figure 4 entropy-25-01105-f004:**
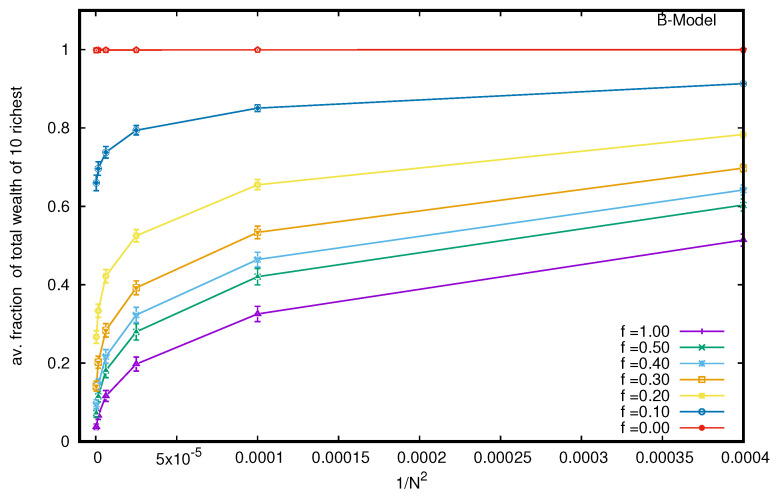
To obtain the limiting values (for large *N*) of the average fraction of total wealth (M=N) possessed by the ten richest traders in the steady state, we plot the fraction against $1/N^{2}$ (as with DY-type trades, each N trader interacts with the *N* − 1 other trader. The extrapolated values all seem to approach zero for any non-zero value of *f* (but there remains a constant 0.9998 for *f* = 0, as in the pure B model). The error estimation is based on 10 runs. Typical sizes of error bars are indicated.

**Figure 5 entropy-25-01105-f005:**
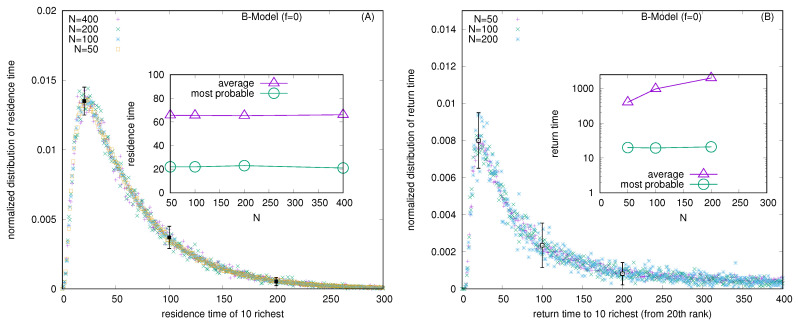
(**A**) The distribution of residence times (in units of *N*) of the 10 fortunate traders and (in the inset) the variation of the most probable and average values of the residence times. (**B**) The distribution of the return time to fortune (becoming one of the 10 richest, starting from the 20th rank) and (in the inset) the variation of the most probable and average values of the return times (in units of *N*). The error estimation is based on 10 runs. The typical errors in the distribution of both the residence and return times grow with *N* near the most probable values of the respective quantities, and are indicated for *N* = 400 here when they are bigger than the symbol sizes.

**Figure 6 entropy-25-01105-f006:**
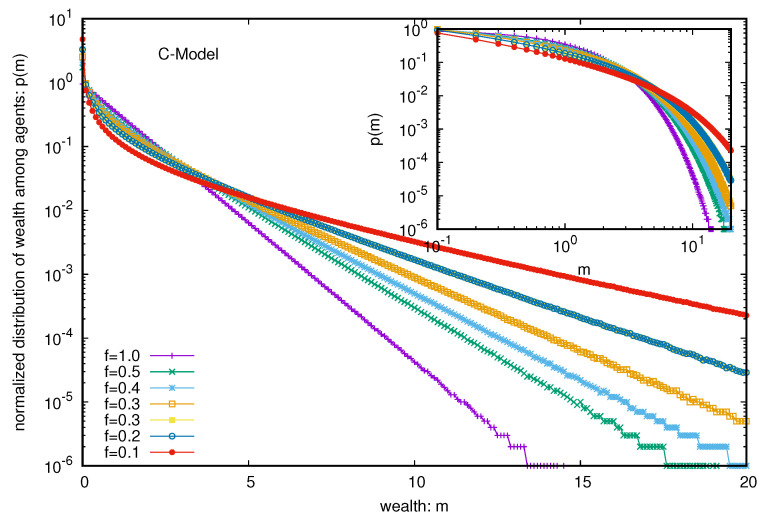
Wealth distribution P(m) among all the agents against the wealth *m* in the C model for different probabilities *f* of DY random exchanges. Note that the fluctuations appear to grow more for the lower values of the distribution due to the log scale used in the y-axis.

**Figure 7 entropy-25-01105-f007:**
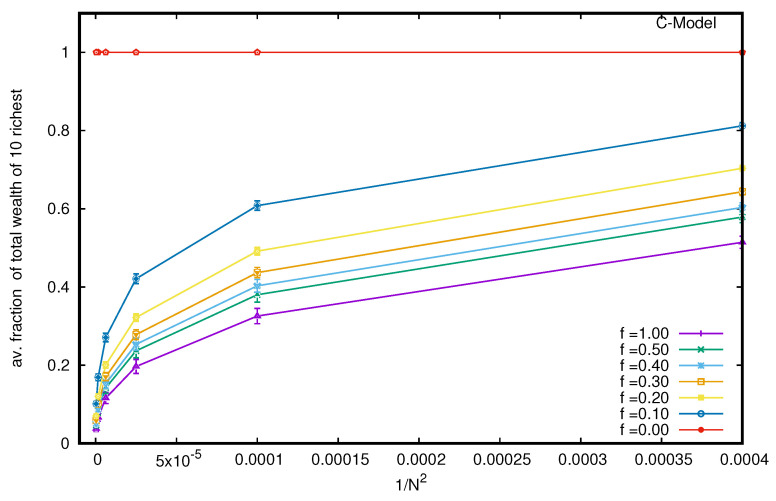
The limiting values (for large *N*) of the average fraction of total wealth (M=N) possessed by the ten richest traders in the steady state of the C model with the *f* fraction of DY-like trades. For f=0, the money goes to one agent and the other nine agents contribute nothing. When we plot the fraction against $1/N^{2}$ (as with DY-type trades, each N trader interacts with (N−1) other traders), the extrapolated values all seem to approach zero for any non-zero value of *f*. The error estimation is based on 10 runs. Typical sizes of error bars are indicated.

**Figure 8 entropy-25-01105-f008:**
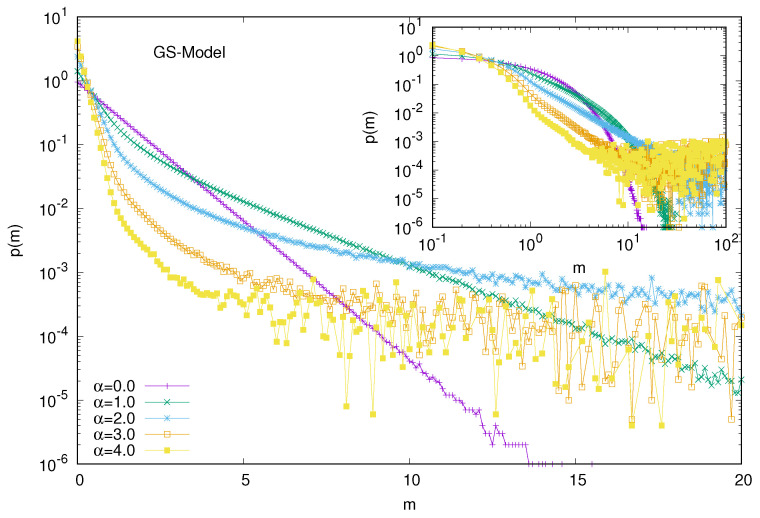
Wealth distribution P(m) among all the agents against the wealth *m* in the GS model for different values of α. Note that the fluctuations appear to grow more for the lower values of the distribution due to the log scale used in the y-axis.

**Figure 9 entropy-25-01105-f009:**
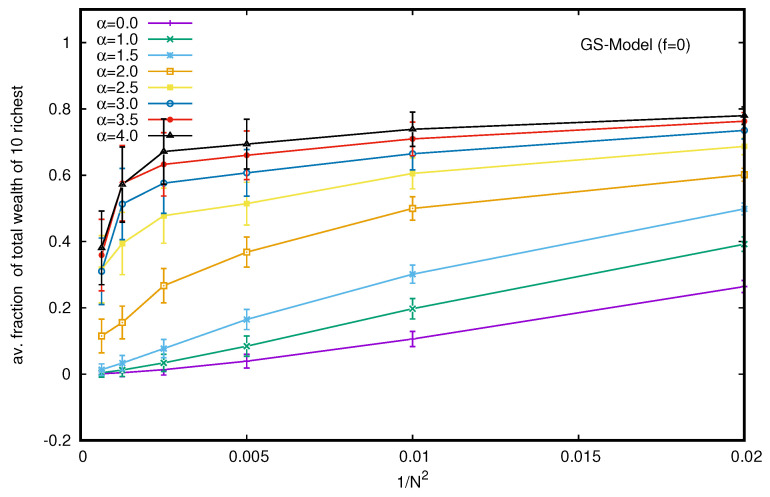
Plot of the fraction of total wealth (M=N) against $1/N^{2}$ for different values of α in the GS model. The extrapolated (with *N*) values of the fraction all seem to approach zero for any non-zero value of α. The error estimation is based on 10 runs. Typical sizes of error bars are indicated.

**Figure 10 entropy-25-01105-f010:**
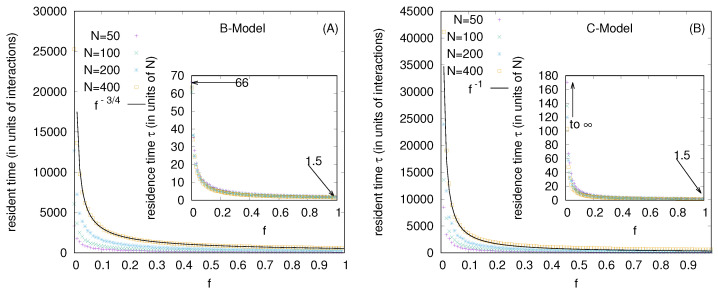
The DY fraction *f* dependence of the bare residence time (in units of interactions or exchanges) at different *N* values for the B-model (**A**) and for the C-model (**B**) are shown. Their power law fits with *f* for *N* = 400 are shown (for other *N* vales, the respective pre-factors change linearly with *N*). The insets show the *f* dependence of the residence times τ (in units of *N*). Note that the limiting values of τ at *f* = 0 are about 66 for the B-model, while they go to infinity for the C-model.

## Data Availability

Code will be available from the corresponding author upon request.

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
