# Peer review of "Kinetic Models of Wealth Distribution with Extreme Inequality: Numerical Study of Their Stability against Random Exchanges"

_entropy, 2023, doi:10.3390/e25071105_

Round 1
Reviewer 1 Report
In this paper, the authors conducted numerical simulations to study the phenomenon of mitigating the dominance of a few individuals by introducing random exchanges with a frequency in three models: the Banerjee model, Chakraborti model, and Goswami-Sen model. The reported results are intriguing, and the observed phenomena seem plausible. However, there are a few points that must be addressed for improvement.
- Specify the number of simulation runs in each figure.
- Show the error-bar in all symbols in all figures.
- Along with the two mentioned points, analyze the significance of the numerical simulation results.
- The current models assume a closed system, but discuss what would happen if they were open systems, where exchanges of energy or particles with the external environment are possible. In my opinion, real economic systems are naturally open systems. Explore the potential implications and differences in the results when considering an open system.
- Currently, the paper presents only simulation results. If possible, add an analytical approach or qualitative interpretation to complement the simulations. This could involve deriving mathematical equations or theoretical explanations to provide a deeper understanding of the observed phenomena.
Author Response
Response to the reviewers' comments:
We are happy to learn that all the three
reviewers appreciated our work: "The
reported results are intriguing, and the
observed phenomena seem plausible"
(Reviewer 1) and "I consider this a very
timely study, carefully conceived and
executed, and likely of interest to many
readers of Entropy" (reviewer 3). We
have revised the paper, following some of
their suggestions and in particular some
additional new results (in the newly added
Figs. 10a,b) have been included.All the
major changes and additions have been
marked in red in this revised version.
Response to the specific suggestions:
A) Reviewer 1:
a) all the simulation details have been
made clear and added.
b) Suggestions for extension to open
systems and for analytical kinetic theory
formulation are appreciated. We hope to
present them in a subsequent paper.
Reviewer 2 Report
the authors should claritfy their motivation of this paper in a more concise way and I recommend them to revise the abstract to improve the readbility.
Author Response
We are happy to learn that all the three
reviewers appreciated our work: "The
reported results are intriguing, and the
observed phenomena seem plausible"
(Reviewer 1) and "I consider this a very
timely study, carefully conceived and
executed, and likely of interest to many
readers of Entropy" (reviewer 3). We
have revised the paper, following some of
their suggestions and in particular some
additional new results (in the newly added
Figs. 10a,b) have been included.All the
major changes and additions have been
marked in red in this revised version.
Response to the specific suggestions:
a) "clarify their motivation of this paper
in a more concise way and I recommend them
to revise the abstract to improve the
readbility."
Thank you & and we have revised the Abstract
as well as the Introduction (and also the
Summary & Discussion section) to clarify our
motivation & the main results reported here
(major changes and additions marked in red).
Reviewer 3 Report
I consider this a very timely study, carefully conceived and executed,
and likely of interest to many readers of Entropy.
The model and methods are clearly explained and the results are sound.
The paper is well written and therefore, I recommend its publication without changes.
Author Response
Response to the reviewers' comments:
We are happy to learn that all the three
reviewers appreciated our work: "The
reported results are intriguing, and the
observed phenomena seem plausible"
(Reviewer 1) and "I consider this a very
timely study, carefully conceived and
executed, and likely of interest to many
readers of Entropy" (reviewer 3). We
have revised the paper, following some of
their suggestions and in particular some
additional new results (in the newly added
Figs. 10a,b) have been included.All the
major changes and additions have been
marked in red in this revised version.
Response to the specific suggestions:
No comment made. Thanks for high appreciation
and recommendation.
Round 2
Reviewer 1 Report
Although the paper has improved slightly compared to the previous one, the results and figures still need further improvement. The suggestions that was raised before but not well addressed are
- Specify the number of simulation runs in each figure.
- Show the error-bar in all symbols in all figures.
- Along with the two mentioned points, analyze the significance of the numerical simulation results.
Author Response
The 1st Reviewer has commented that
"Although the paper has improved
slightly compared to the previous one,
the results and figures still need
further improvement." and suggested
a few specific points on which the
presentation "Can be improved".
Our responses on those specific points
are:
> Specify the number of simulation
> runs in each figure:
Each simulation run goes for 10^5
iterations/trades after the steady
state is achieved as mentioned already
in section2, see lines 133-137.
> Show the error-bar in all symbols in
> all figures:
Typical error-bars are estimated from
10 runs and are indicated in all figures
wherever they are more than the symbol sizes.
> Along with the two mentioned points,
> analyze the significance of the
> numerical simulation results.
Numerical observation of the growth of
errors/fluctuations near the most
probable values of wealth (Figs 1, 2) and
residence/return times (Fig. 5) in the
B-model for $f = 0$) is noted and briefly
mentioned in the summary and Discussion
section (line nos. 230-233).
Hope, the paper is now acceptable for
publication.